# Can Environmental Regulation Reduce Urban Haze Concentration from the Perspective of China's Five Urban Agglomerations?

**Xinfei Li** [1], **Yueming Li** [1], **Chang Xu** [2], **Jingyang Duan** [1], **Wenqi Zhao** [1], **Baodong Cheng** [1,*] and **Yuan Tian** [3,*]

[1]  School of Economics and Management, Beijing Forestry University, Beijing 100083, China; xinfeili@bjfu.edu.cn (X.L.); lym_aria98@bjfu.edu.cn (Y.L.); duanjingyang0526@bjfu.edu.cn (J.D.); zhaowq@bjfu.edu.cn (W.Z.)
[2]  School of Finance and Public Management, Anhui University of Finance and Economics, Bengbu 233030, China; xuchang@bjfu.edu.cn
[3]  School of Business, Beijing Union University, Beijing 100025, China
*  Correspondence: baodong@bjfu.edu.cn (B.C.); yuan.tian@buu.edu.cn (Y.T.)

**Abstract:** Based on the perspective of urban agglomerations, this paper explores the impact mechanism of environmental regulation on haze, and tries to find the most suitable environmental regulation intensity for haze control in urban agglomerations. This paper uses the fixed-effect model and panel threshold model to verify the effect of environmental regulations on haze concentration in 206 cities in China. A grouping test is also conducted to verify whether a regional heterogeneity arises due to different regional development levels for five urban agglomerations and non-five urban agglomerations, respectively. The results show that: (1) In the linear model, strengthening environmental regulation can reduce the haze concentration, but this effect is not significant. The effect of environmental regulation on haze control in the five major urban agglomerations is better than that in the non-five major urban agglomerations; (2) In the nonlinear model, the impact of environmental regulation on haze shows a "U" trend in the five major urban agglomerations and an inverted "U" trend in the non-five major urban agglomerations. Although the results are not significant, we can still conclude that the impact of environmental regulation on haze varies depending on the level of regional economic development. Therefore, the environmental regulation should be formulated according to local conditions; (3) In the threshold model, the impact of environmental regulation on the haze concentration in five major urban agglomerations has a threshold effect. In the five major urban agglomerations, although environmental regulation can effectively reduce haze concentration, the governance effect will weaken as the environmental regulation increases. This study plays a positive role in guiding local governments to adjust environmental regulation intensity according to local conditions and helping local environmental improvement.

**Keywords:** environmental regulation; haze concentration; heterogeneity

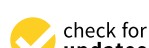

## 1. Introduction

With the rapid development of China's economy, the ecological environment has become increasingly severe due to excessive energy consumption, and the haze pollution has gradually increased. This has seriously affected the health of urban residents. Therefore, haze pollution has become the biggest threat to public physical and mental health [1–4]. Severe haze pollution not only seriously threatens people's health, but also hinders economic development [5,6]. For sustainable economic and social development, China began to follow the example of neighboring developed countries to try control pollution through effective environmental regulation [7,8].

Urban agglomerations are the backbone to promote the high-quality development of regional economy. The five major urban agglomerations include Pearl River Delta Urban Agglomeration, Middle Reaches of the Yangtze River Urban Agglomeration, Yangtze

River Delta Urban Agglomeration, Bohai Bay Urban Agglomeration, and Cheng Yu Urban Agglomeration [9]. The economic scale and innovation capacity of the five urban agglomerations are far ahead of other regions in China. Behind this achievement, there are inevitable phenomena such as high-density urban construction, high-intensity consumption of resources, and high industrial agglomeration development, which result in huge hidden dangers such as ecological damage and environmental pollution [10,11]. Existing studies believe that there is regional heterogeneity in the impact of environmental regulation on haze in China, which is reflected not only in the difference in urban geographical location [12], but also in the disparities in economic development levels [13–15]. Since the economic development level of the five urban agglomerations is significantly better than that of the non- five urban agglomerations, we speculate that the impact of environmental regulation on haze would also be different from that of other regions. Based on this, this paper analyzes the effect of environmental regulation on haze concentration from the perspective of five major urban agglomerations using a fixed-effects model and a threshold model to verify whether the effect of environmental regulation on haze concentration is regionally heterogeneous due to different regional economic levels.

At present, scholars mainly debate whether environmental regulation can significantly suppress haze pollution. The supporting view is that environmental regulation is conducive to improving the environmental quality and controlling haze concentration [16–19]. This is mainly based on the "Forced emission reduction". Specifically, by setting higher emission standards, the government urges enterprises to improve production levels and pollution control technologies to avoid high pollution penalties and realize pollution reduction [20,21]. Lorente et al. [22] found that cross-regional environmental regulations have remarkable addressing haze pollution. Levinsohn and Petrin [23] conducted an empirical analysis based on the data from the American paper industry and found that high-intensity environmental regulation facilitated the prevention of haze pollution. Wang [24] and Sigman [25] also came to a similar conclusion, that is, environmental regulation is conducive to controlling haze and improving environmental quality. The opposing view is the "Green paradox". With unchanged consumer demand and production technology, increased environmental regulation would make enterprises reduce the cost of technological innovation because of increased environmental treatment cost. The high costs hinder improvements in production and sewage technologies. This will cause more air pollutants emissions [26], leading to local environmental deterioration and aggravating the haze pollution throughout the region [27–29]. Smulders et al. [30] confirmed this view that strict environmental regulation significantly exacerbated environmental pollution. Greenstone [30] pointed out that strict environmental regulations inhibit the scientific and technological innovation of American enterprises and discourage pollution-intensive firms from reducing emissions. In addition, some scholars have different views on the effect of environmental regulation on haze. Some scholars believe that the impact of environmental regulation on haze cannot be determined [31]. Others believe that the effect of environmental regulation is influenced by the level of local economic development [32]. Nesadurai [33] considered the economic development level of different regions and the impact of haze pollution from multiple levels and angles. They proposed that it is important to plan for local conditions and develop haze control policies with different regional priorities.

Although the existing studies have proposed that there is regional heterogeneity in the treatment effect of environmental regulation on environmental pollution, these studies are still insufficient. Firstly, the literature analyzing the impact effect of environmental regulation mostly selects China's provincial panel data. The research results are not accurate compared with municipal units, and there is likely to be errors [15]. Secondly, although studies have proposed that the impact of environmental regulation on China's haze would vary according to the regional economic level, the regions are mostly divided into east, middle, and west regions [14]. This way of dividing economic regions is not as obvious as the economic difference between urban agglomeration and non-urban agglomeration. Finally, in order to more accurately measure the effect of urban environmental regulation,

this paper selects five environmental governance indicators, namely industrial SO$_2$ removal rate, soot removal rate, industrial solid waste comprehensive utilization rate, domestic sewage treatment rate, and domestic garbage harmless treatment rate. Based on the five environmental governance indicators, this paper uses the entropy weight method to calculate the intensity of environmental regulation, which makes the quantification of environmental regulation more comprehensive and persuasive.

For practical situations, this paper attempts to answer the following questions: (1) Does China's implementation of environmental regulation have a significant impact on haze concentration? (2) Is the impact of environmental regulation on haze concentration positive, negative, or nonlinear? (3) The economic development and haze pollution of the five urban agglomerations are higher than those in other regions. Is the effect of environmental regulation on controlling haze in the five urban agglomerations different from that in other regions? Reasonable answers to the above questions not only help to improve the relevant literature research on environmental regulation, but also play a positive role in guiding local governments to adjust the strength of environmental regulation according to local conditions and help local environmental improvement. To a certain extent, it also promotes the sustainable development of society and has important theoretical and practical significance for the economic transformation of Chinese society.

## 2. Environmental Regulation Intensity

According to the theory of resource and environmental economics, resources are scarce. The use of environmental resources by one party may have negative externalities to other users. For example, the economic activities of pollutant discharge enterprises can bring damage to nearby residents or other enterprises, generate social costs, and lead to inefficient use of environmental resources. In the early 20th century, in order to overcome the inefficient use of environmental resources, the government took various measures and formed the prototype of environmental regulation.

In 1980, Dasgupta first introduced the concept of environmental regulation. He believed that environmental regulation is only the push-pull effect of government policies, that is, the policies and mandatory means formulated by the government to ensure economic development while taking into account the ecological environment, so as to reduce the external diseconomy caused by pollutant emissions [34]. With the development of the times, it is generally believed that environmental regulation is a binding force with the purpose of environmental protection, the object of individual or organization, and the form of a tangible system or intangible consciousness. At present, the purpose of environmental regulation is to improve environmental quality, accelerate the transformation of economic growth mode and promote the process of industrial structure upgrading [35].

In the current study, the measurement of environmental regulation is not uniform, mainly including two types: one is to use different pollutant emission densities [36,37]. The other is to use environmental regulation policies [38,39]. Because the above indicators are relatively single and insufficient to represent environmental regulation, this paper adopts the entropy weight method to construct a comprehensive measurement system of urban environmental regulation.

This paper is based on the five indices of industrial SO$_2$ removal rate, soot removal rate, industrial solid waste comprehensive utilization rate, domestic sewage treatment rate, and domestic garbage harmless treatment rate provided by "China City Statistical Yearbook" (2007–2017) and "China Regional Economic Statistical Yearbook" (2007–2017). This paper obtains the municipal unit environmental regulation intensity index by standardization to gain its entropy value.

The first step is to standardize the raw data.

$$P''_{ij} = \frac{X_{ij} - min(X_{ij})}{\max(X_{ij}) - min(X_{ij})}. \tag{1}$$

Among them, $X_{ij}$ represents the value of the $j$ environmental pollution index of $i$ city. The second step is to perform coordinate translation on the standardized data.

$$P'_{ij} = 1 + P''_{ij}. \tag{2}$$

The third step is to calculate the proportion of the $j$ environmental pollution index in the $i$ city.

$$P_{ij} = P'_{ij} / \sum_{i=1}^{m} P'_{ij}. \tag{3}$$

The fourth step is to calculate the entropy and coefficient of variation of the $j$ environmental pollution index.

$$e_j = \left( \frac{1}{\ln m} \right) \sum_{i=1}^{m} P_{ij} \ln (P_{ij}), \tag{4}$$

$$g_j = 1 - e_j. \tag{5}$$

The fifth step is to calculate the weight of the $j$ environmental pollution index in the comprehensive evaluation.

$$W_j = g_j / \sum_{j=1}^{n} g_j \tag{6}$$

The sixth step is to calculate the comprehensive index of environmental pollution.

$$\mathrm{GER}_i = \sum_{j=1}^{n} W_j P_{ij}. \tag{7}$$

The $\mathrm{GER}_i$ represents the intensity of environmental regulation. The greater the value of $\mathrm{GER}_i$, the greater the intensity of the city's implementation of environmental regulations.

## 3. Research Model and Method

### 3.1. Data Sources

The research object of this paper is prefecture-level and above-level cities in China. Due to the continuity and availability of data, the selected sample is 206 prefecture-level and above cities from 2006 to 2016, with a total of 2266 sample values. Among them, 82 cities are in five major urban agglomerations (Table 1), and 124 cities are in non- five urban agglomerations. We have marked the location of the five urban agglomerations in Figure 1. The data of computing environment regulation are mainly derived from the "China City Statistical Yearbook" (2007–2017) and "China Regional Economic Statistical Yearbook" (2007–2017). The variable to measure haze concentration is PM2.5, which comes from the Colombian University Socioeconomic Data and Applications Center (SEDAC).

**Table 1.** The cities included in the five urban agglomerations.

| Urban Agglomeration | City |
|---|---|
| Pearl River Delta Urban Agglomeration (19) | Guangzhou Shaoguan Shenzhen Zhuhai Shantou Foshan Jiangmen Zhaijiang Meizhou Maoming Huizhou Shanwei Heyuan Yangjiang Qingyuan Zhaoqing Dongwan Zhongshan Chaozhou |
| Middle Reaches of the Yangtze River Urban Agglomeration (20) | Pingxiang Xinyu Yingtan Wuhan Huangshi Shiyan Yichang Ezhou Jingmen Xiaogan Jinzhou Huanggang Changsha Zhuzhou Xiangtan Hengyang Shaoyang Yueyang Changde Yiyang |
| Yangtze River Delta Urban Agglomeration (21) | Nanjing Wuxi Xuzhou Changzhou Suzhou Nantong Lianyungang Huainan Yancheng Yangzhou Zhenjiang Hangzhou Ningbo Wenzhou Jiaxing Huzhou Shaoxing Jinhua Quzhou Zhoushan Taizhou |

**Table 1.** *Cont.*

| Urban Agglomeration | City |
|---|---|
| Bohai Bay Urban Agglomeration (11) | Shijiazhuang Tangshan Qinhuangdao Handan Xingtai Baoding Zhangjiakou Chengde Cangzhou Langfang Hengshui |
| Cheng Yu Urban Agglomeration (11) | Chengdu Zigong Paizhihua Luzhou Deyang Mianyang Guangyaun Suining Neijiang Leshan Nanchong |

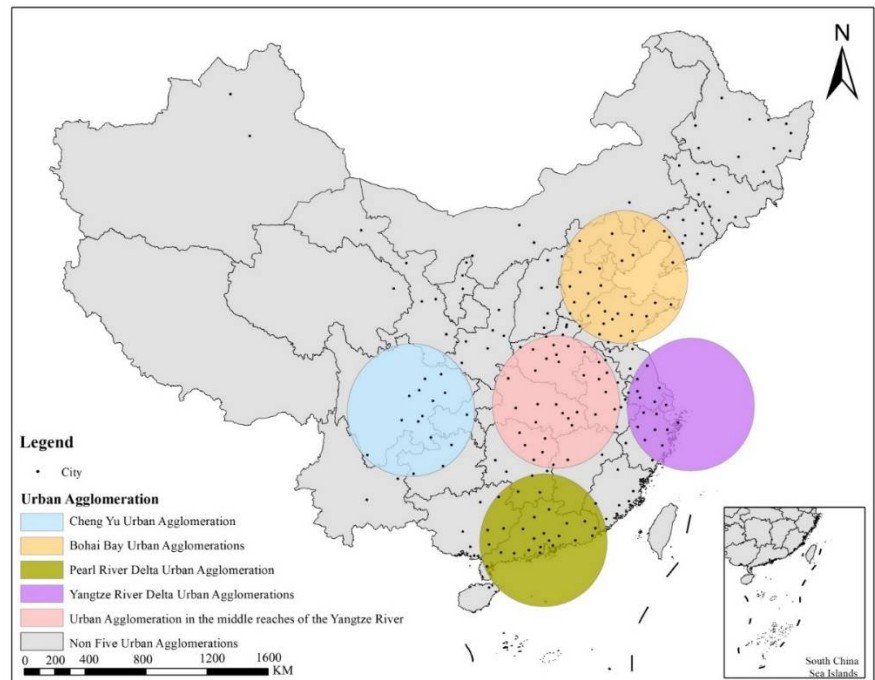

**Figure 1.** Location of five urban agglomerations in China.

### *3.2. Model*

In order to investigate the impact of environmental regulation on the haze control, we used the fixed-effect model to analyze the urban data based on the Hausmann test results. When constructing the model, this paper not only considered the variables in general sense, but also included the quadratic term ($GER^2$) of environmental regulation (GER) to analyze the nonlinear impact of environmental regulation on haze under the fixed-effect. The model is as follows:

$$PM2.5_{i,t} = \alpha_0 + \alpha_1 GER_{i,t} + \alpha_n CONTROLS_{i,t} + \sum YearDummy + \varepsilon_{i,t}, \tag{8}$$

$$PM2.5_{i,t} = \alpha_0 + \alpha_1 GER_{i,t} + \alpha_2 GER^2{}_{i,t} + \alpha_n CONTROLS_{i,t} + \sum YearDummy + \varepsilon_{i,t}. \tag{9}$$

Models (1) and (2) are used to test the linear and nonlinear impact of environmental regulation on haze concentration, where $PM2.5_{i,t}$ represents urban haze pollution. In addition, $\varepsilon_{i,t}$ is the control variable and is the random disturbance term. $\sum YearDummy$ indicates the control time effect. Where $i$ and $t$ denote individual cities ($i$ = 1, 2, . . . , 206) and time ($t$ = 2006, 2004, . . . , 2016), respectively.

### *3.3. Variable Selection*

The explained variable is PM2.5 to measure haze concentration. There are two methods for measuring haze concentration in the existing literature. First, some scholars use the Air Quality Index (AQI) released by China. However, AQI is a comprehensive statistic

on various air pollutants such as sulfur dioxide, nitrogen oxides, and inhalable particulate matter. Therefore, it is impossible to measure haze pollution accurately. Second, other scholars select the relevant data of Columbia University socioeconomic data and Applications Center (SEDAC). This is based on the global PM2.5 concentration data extracted by Donkelaar et al. [40], which is currently updated to 2016. The haze is mainly composed of sulfur dioxide, nitrogen oxides and inhalable particles. Sulfur dioxide and nitrogen oxides are gaseous pollutants, and inhalable particulate matter is the main contributor to haze weather pollution. They combine with fog to make the sky gloomy and gray. Therefore, this paper selected PM2.5 as the variable to measure the severity of haze. The data source is Columbia University socioeconomic data and Applications Center (SEDAC). This data is based on satellite monitoring compared to ground-based field detection data. Therefore, this database can reflect the changes of PM2.5 concentration more comprehensively and accurately.

The explanatory variable of this paper is the environmental regulation intensity (GER). In this paper, the industrial $SO_2$ removal rate, soot removal rate, industrial solid waste comprehensive utilization rate, domestic sewage treatment rate, and domestic garbage harmless treatment rate was selected, and the entropy value is calculated by standardizing them to obtain the environmental regulation intensity index.

In order to make the econometric model more robust and reduce the estimation errors that may be brought by omitted variables, the control variables and their measures selected in this paper are shown in Table 2.

**Table 2.** Variable selection.

| Classification | Name | Interpretation | Symbol | Ref. |
|---|---|---|---|---|
| Explained variable | Haze concentration | PM2.5 concentration data is based on the grid data of global PM2.5 concentration from 2003 to 2016 provided by the social and economic data and application center of Columbia University | PM2.5 | [41] |
| Explanatory variable | Environmental regulation intensity | The intensity of environmental regulation is calculated by entropy weight method through the five single indices of industrial $SO_2$ removal rate, soot removal rate, industrial solid waste comprehensive utilization rate, domestic sewage treatment rate, and domestic garbage harmless treatment rate. | GER | [42] |
| Control | Density of population | Total population/administrative land area | PEO | [43] |
| | Level of education | Number of college students/total urban population | ST | [44] |
| | Level of urban development | GDP growth rate = (GDP of the previous year − GDP of the current year)/GDP of the previous year | GDP | [45] |
| | Industrial structure | Added value of tertiary industry/added value of secondary industry | IS | [46] |
| | Opening up | Total industrial Output value of foreign-invested enterprises (CNY 10,000)/Gross regional Product (CNY 10,000) | FDI | [47] |
| | Infrastructure | Urban road area per capita | ROD | [48] |
| | Government education input | The natural logarithm of education expenditure (CNY 10,000) | ED | [14] |
| | Government R&D (Research and Development) investment | The natural logarithm of research and development expenditure (CNY 10,000) | RD | [49] |

## 4. Analysis of Empirical Results

### 4.1. Data Description

The results of annual changes in the environmental regulation intensity in Figure 2 show that the overall intensity of environmental regulation tends to increase, except for a slight decline in 2011 and 2012. The intensity of environmental regulation in the five urban agglomerations is consistently higher than that in non- five urban agglomerations.

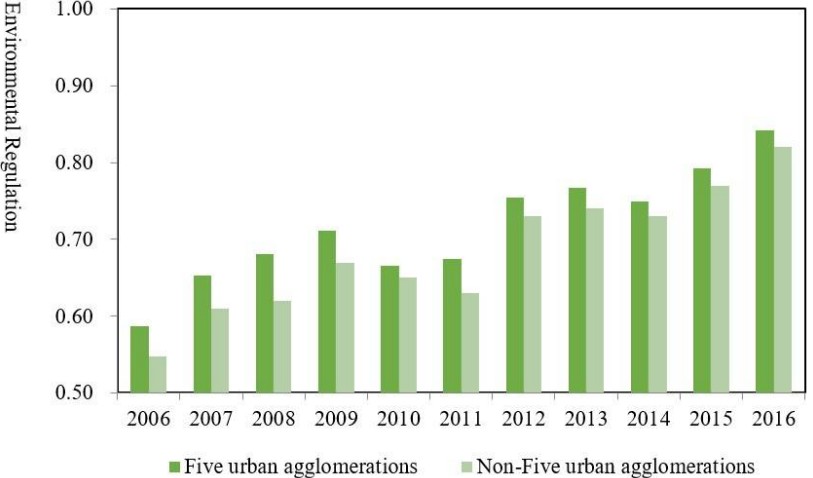

**Figure 2.** Changes in environmental regulation intensity of five major urban agglomerations and non-five major urban agglomerations.

As shown in Figure 3, the same grouping method is used for the changes in PM2.5. The results show that PM2.5 concentration shows an overall decreasing trend, and PM2.5 concentration of the five urban agglomerations is consistently higher than that of the non-five urban agglomerations.

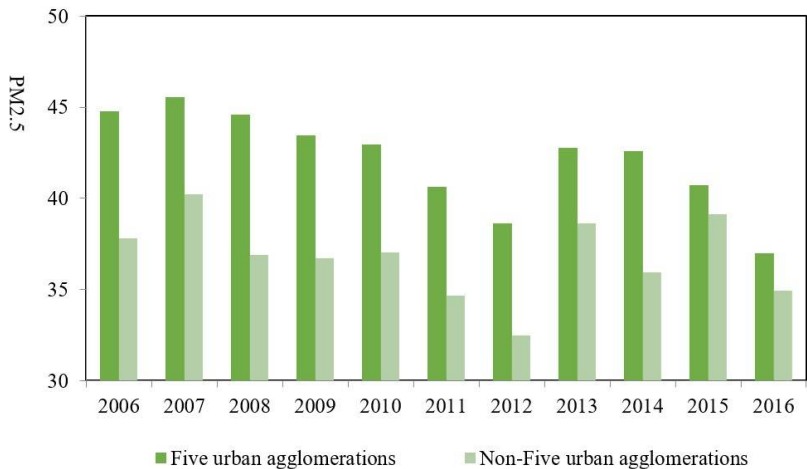

**Figure 3.** Changes in PM2.5 of five major urban agglomerations and non-five major urban agglomerations.

### 4.2. Panel Fixed Model

The linear effects of environmental regulation on haze concentration are shown in Table 3. The research object of model (1) is 206 cities in China, while models (2) and (3) are respectively aimed at 82 cities located in the 5 major urban agglomerations and 124 cities in non-5 major urban agglomerations. The linear model shows that the impact of environmental regulation (GER) on haze concentration (PM2.5) is always negative, despite the different samples according to the degree of economic development. In other words,

the implementation of environmental regulation can reduce the haze concentration, but this effect is not significant. By comparing the coefficients of models (2) and (3), it can be seen that the effect of environmental regulation (GER) on haze concentration (PM2.5) in the five major urban agglomerations is better than that in the non-five major urban agglomerations.

**Table 3.** Linear models of national, five major urban agglomerations and non-five major urban agglomerations.

| Variables | Nationwide | Five Urban Agglomerations | Non-Five Urban Agglomerations |
| --- | --- | --- | --- |
| | Model (1) | Model (2) | Model (3) |
| GER | −0.710 | −1.655 | −0.933 |
| | (1.122) | (1.648) | (1.474) |
| PEO | 0.001 | −0.001 | 0.002 |
| | (0.001) | (0.001) | (0.001) |
| ST | −0.001 * | −0.001 | −0.002 |
| | (0.001) | (0.001) | (0.001) |
| IS | 1.157 *** | 0.824 | 0.659 |
| | (0.419) | (0.765) | (0.508) |
| FDI | −1.830 | −1.508 | −5.378 *** |
| | (1.123) | (1.315) | (1.886) |
| ROD | 0.004 | −0.0595 | 0.023 |
| | (0.027) | (0.042) | (0.033) |
| ED | −1.467 *** | −0.385 | −1.746 *** |
| | (0.368) | (0.645) | (0.442) |
| RD | −0.648 *** | 0.306 | −0.727 *** |
| | (0.175) | (0.306) | (0.212) |
| GDP | −0.006 | −0.005 | −0.007 |
| | (0.008) | (0.009) | (0.013) |
| Constant | 60.06 *** | 49.32 *** | 60.97 *** |
| | (3.713) | (6.684) | (4.383) |
| Observations | 2266 | 902 | 1364 |
| Number | 206 | 82 | 124 |
| R-squared | 0.263 | 0.371 | 0.272 |

Note: * and *** respectively represent that the estimated coefficient is significant at the confidence levels 10% and 1%, and the standard error of the coefficient is marked in parentheses.

In the linear model, although environmental regulation (GER) can reduce the haze concentration (PM2.5), the effect is not significant. We consider whether the impact of environmental regulation (GER) on haze concentration (PM2.5) would show a significant nonlinear impact. The results of the nonlinear impact of environmental regulation (GER) on haze concentration (PM2.5) are shown in Table 4. Models (4)–(6) are based on the different samples. Among them, model (4) is based on a sample of 206 cities in China. The model (5) is based on a sample of 82 cities located in 5 major urban agglomerations. The model (6) is based on a sample of 124 cities located in non-5 major urban agglomerations.

The impact of environmental regulation (GER) in models (4) and (6) is basically similar, showing an inverted "U" trend. Specifically, the primary term (GER) of environmental regulation has a positive impact on haze concentration, while the secondary term (GER$^2$) of environmental regulation has a negative impact on haze concentration (PM2.5), which is an inverted "U" curve. This shows that whether the research samples consist of 206 cities in China or 124 cities in non-5 major urban agglomerations, the implementation of environmental regulation would increase the haze concentration first and then decrease with the improvement of the intensity of environmental regulation. Although the trend is the same, the critical points of models (4) and (6) are different. The critical point in model (4) is marked as 0.48 ($\frac{-\text{GER}}{-2 \times \text{GER}^2} = \frac{2.240}{2 \times 2.329}$), and the critical point in model (6) is 0.54 ($\frac{-\text{GER}}{-2 \times \text{GER}^2} = \frac{5.991}{2 \times 5.535}$). For 206 cities in China, when the environmental regulation intensity is less than 0.48, the increase of environmental regulation intensity would increase the haze concen-

tration. When the environmental regulation intensity is greater than 0.48, the increase of environmental regulation intensity would reduce the haze concentration. For 124 cities in non-5 major urban agglomerations, when the environmental regulation intensity is less than 0.54, the increase of environmental regulation intensity would increase the haze concentration. When the environmental regulation intensity is greater than 0.54, the increase of environmental regulation intensity would reduce the haze concentration.

Contrary to the results of models (4) and (6), model (5) reports the governance effect of environmental regulation (GER) on haze concentration (PM2.5) in five urban agglomerations. According to model (5) for 82 cities in 5 urban agglomerations, the impact of environmental regulation (GER) on haze concentration (PM2.5) shows a "U" shape change. The critical point in five urban agglomerations is marked as 0.75 ($\frac{-GER}{-2 \times GER^2} = \frac{11.64}{2 \times 7.743}$). Specifically, when the intensity of environmental regulation is less than 0.75, the increase of environmental regulation intensity would reduce the haze concentration. When the intensity of environmental regulation is greater than 0.75, the increase of environmental regulation intensity would increase the haze concentration.

Therefore, an intensity of environmental regulation less than 0.75 is suitable to reduce haze concentration for the five urban agglomerations. On the contrary, an intensity of environmental regulation greater than 0.54 is suitable to reduce haze concentration for the non-five urban agglomerations. Although the results are not significant, we can still see that the impact of environmental regulation on haze would be different due to the level of regional economic development. Therefore, the intensity of environmental regulation should be formulated according to local conditions.

**Table 4.** Nonlinear models of national, five major urban agglomerations, and non-five major urban agglomerations.

| Variables | Nationwide | Five Urban Agglomerations | Non-Five Urban Agglomerations |
|---|---|---|---|
| | Model (4) | Model (5) | Model (6) |
| GER | 2.240 | −11.640 | 5.991 |
| | (5.438) | (8.415) | (6.885) |
| GER$^2$ | −2.329 | 7.743 | −5.535 |
| | (4.200) | (6.397) | (5.377) |
| PEO | 0.001 | −0.001 | 0.001 |
| | (0.001) | (0.001) | (0.001) |
| ST | −0.001 * | −0.001 | −0.002 |
| | (0.001) | (0.001) | (0.001) |
| IS | 1.167 *** | 0.792 | 0.690 |
| | (0.420) | (0.765) | (0.509) |
| FDI | −1.857 * | −1.407 | −5.409 *** |
| | (1.124) | (1.317) | (1.886) |
| ROD | 0.003 | −0.062 | 0.022 |
| | (0.027) | (0.042) | (0.033) |
| ED | −1.473 *** | −0.367 | −1.765 *** |
| | (0.368) | (0.645) | (0.442) |
| RD | −0.635 *** | 0.285 | −0.688 *** |
| | (0.176) | (0.306) | (0.215) |
| GDP | −0.006 | −0.005 | −0.007 |
| | (0.008) | (0.009) | (0.013) |
| Constant | 59.19 *** | 52.23 *** | 58.96 *** |
| | (4.036) | (7.100) | (4.801) |
| Observations | 2266 | 902 | 1364 |
| Number | 206 | 82 | 124 |
| R-squared | 0.263 | 0.373 | 0.273 |

Note: * and *** respectively represent that the estimated coefficient is significant at the confidence levels 10% and 1%, and the standard error of the coefficient is marked in parentheses.

*4.3. Threshold Model Analysis*

The fixed-effect shows a nonlinear impact of environmental regulation on haze, but the results are not significant. Therefore, we choose the threshold model for further verification. The threshold model is the synthesis of several simple linear models. Compared with the general nonlinear model, the model has stronger explanatory power [30]. Environmental regulation is used as the threshold variable to reveal the pattern and threshold characteristics of environmental regulation on haze concentration. We set the following panel threshold regression model for testing. The specific form is as follows:

$$PM2.5_{i,t} = c + \beta_1 GER_{i,t} I(GER_{i,t} \leq \gamma_1) + \beta_2 GER_{i,t} I(\gamma_1 < GER_{i,t} < \gamma_2) + \cdots\cdots + \beta_n GER_{i,t} I(GER_{i,t} > \gamma_n) + \theta X_{i,t} + u_i \quad (10)$$

$X_{i,t}$ are control variables in addition to the variables explained above. $e_{i,t}$ is the random disturbance term; $u_i$ is the individual effect. $I(\cdot)$ is the indicative function, and the function takes the value of 1 when the condition in parentheses holds, and 0 otherwise. $\gamma$ is the threshold value to be estimated; $\beta_1 \sim \beta_n$ denotes the elasticity coefficients of the threshold variables on PM2.5 in different zones, respectively, and the existence of the threshold effect is judged by testing whether the estimated values or signs of $\beta_1 \sim \beta_n$ exhibit significant differences.

The economic development level of the five major urban agglomerations is higher than that of the non-five major urban agglomerations. Some literatures [34] have proposed that the impact of environmental regulation on haze concentration would be influenced by the regional economy. Therefore, it is inferred that there are regional differences in the impact of environmental regulation and economy on haze in China. The panel linear model and nonlinear model have verified this view, but the effect is not significant. Therefore, we use a more accurate threshold model to further test the impact of environmental regulation on haze concentration in different regions. In the paper, the corresponding p-value and confidence intervals are obtained by bootstrap sampling 300 times [50]. It can be seen from Table 5 that the threshold test results reject the triple, double, and single thresholds in the national and non-five major urban agglomerations. Therefore, the model based on these two samples has no threshold effect. Substituting the five urban agglomerations, the double threshold model passed the test. The threshold values are 0.49 and 0.59, respectively, that is, the intensity of environmental regulation is divided into three intervals, which are low level (GER ≤ 0.49), medium level (0.49 < GER ≤ 0.59), and high level (GER > 0.59). The threshold regression results are shown in Table 6.

**Table 5.** Threshold effect test of environmental regulation.

| Sample | Model | Threshold | F-Statistic (F) | p-Value (p) | Bootstrap (BS) |
|---|---|---|---|---|---|
| Nationwide (206) | Single threshold | 0.71 | 5.83 | 0.37 | 300 |
| | Double threshold | 0.71 0.73 | 2.52 | 0.84 | 300 |
| | Triple threshold | 0.69 0.71 0.73 | 5.6 | 0.29 | 300 |
| Five Urban Agglomerations (82) | Single threshold | 0.86 | 6.72 | 0.33 | 300 |
| | Double threshold | 0.49 0.59 | 10.32 ** | 0.06 | 300 |
| | Triple threshold | 0.49 0.59 1.01 | 6.21 | 0.38 | 300 |
| Non-Five Urban Agglomerations (124) | Single threshold | 0.74 | 6.07 | 0.36 | 300 |
| | Double threshold | 0.71 0.74 | 4.09 | 0.49 | 300 |
| | triple threshold | 0.69 0.71 0.74 | 6.07 | 0.47 | 300 |

Note: ** represents that the estimated coefficient is significant at the confidence levels 5%, and the standard error of the coefficient is marked in parentheses.

The sample of the five urban agglomerations passed the threshold test and was identified as a double threshold model. The threshold regression results of environmental regulation of the five urban agglomerations are shown in Table 6. GER-1 indicates low intensity of environmental regulation (GER ≤ 0.49). GER-2 indicates medium intensity of environmental regulation (0.49 < GER ≤ 0.59). GER-3 indicates high intensity of environmental regulation (GER > 0.59). Model (7) is the current data of five urban agglomerations. The impact of low intensity environmental regulation (GER ≤ 0.49) on haze concentration is significantly negative at the level of 1%, and the coefficient is −12.04, that is, the implementation of environmental regulation is conducive to reducing haze concentration. The impact of medium intensity environmental regulation (0.49 < GER ≤ 0.59) on haze concentration is also significantly negative at the level of 1%. The coefficient is −7.13, which means that although the implementation of environmental regulation is conducive to reducing haze concentration, the effect in this phase is slightly less effective than that in the first phase. At higher intensity of environmental regulation (GER > 0.59), the impact of environmental regulation on haze concentration is still significant at 1%, and the coefficient is 5.23, indicating that when the intensity of environmental regulation increases, although haze concentration always shows a significant downward trend, the impact is also weakening.

To test the robustness of the model results, this paper deals with the environmental regulation variables with lag period 1 and period 2. According to the threshold effect test, the models with lag period 1 and lag period 2 do not pass the double and triple threshold tests, but pass the single threshold test. The model with lag period 1 is shown in model (8), with a single threshold of 0.83. The model with 1 lag period 2 is shown in model (9) with a single threshold of 0.77. Thus, the environmental regulation intensity of model (8) is divided into two distinctions: low level (GER ≤ 0.83) and high level (GER > 0.83). Therefore, the environmental regulation intensity of model (9) is divided into two distinctions: low level (GER ≤ 0.77) and high level (GER > 0.77). In model (8) with lag period 1, the impact of low intensity environmental regulation (GER ≤ 0.83) on haze concentration is significantly negative at the level of 1%, and the coefficient is −4.64. The implementation of environmental regulation at this stage is conducive to reducing haze concentration. At higher haze concentration (GER > 0.83), although environmental regulation would reduce haze concentration, the effect is not significant. In model (9) with lag period 2, the impact of low intensity environmental regulation (GER ≤ 0.77) on haze concentration is significantly negative at the level of 1%, and the coefficient is −6.07. At this stage, the implementation of environmental regulation is conducive to reducing haze concentration. At higher haze concentration (GER > 0.77), the impact on haze concentration is significantly negative at the level of 1%, and the coefficient is −8.117. At this stage, the implementation of environmental regulation would further significantly reduce haze concentration, and the effect is stronger than that of environmental regulation range (GER ≤ 0.77).

**Table 6.** Threshold regression results of environmental regulation in five urban agglomerations.

| Variables | Original Model | Lag Period 1 | Lag Period 2 |
|---|---|---|---|
| | Model (7) | Model (8) | Model (9) |
| GER-1 | −12.04 *** | −4.638 *** | −6.070 *** |
| | (3.163) | (1.687) | (1.713) |
| GER-2 | −7.131 *** | −2.630 | −8.117 *** |
| | (2.141) | (1.599) | (1.569) |
| GER-3 | −5.233 *** | | |
| | (1.859) | | |
| PEO | 0.004 *** | 0.004 *** | 0.004 *** |
| | (0.001) | (0.001) | (0.001) |
| ST | −0.002 ** | −0.001 | 0.0001 |
| | (0.001) | (0.001) | (0.001) |

**Table 6.** *Cont.*

| Variables | Original Model | Lag Period 1 | Lag Period 2 |
|---|---|---|---|
| | Model (7) | Model (8) | Model (9) |
| IS | −0.221 | 0.0312 | 0.507 |
| | (0.612) | (0.815) | (0.839) |
| FDI | −0.856 | −1.560 | −3.482 ** |
| | (1.214) | (1.526) | (1.620) |
| ROD | −0.0928 * | −0.0990 ** | −0.0781 * |
| | (0.0474) | (0.0479) | (0.0455) |
| ED | −3.097 *** | −1.985 *** | −1.112 ** |
| | (0.460) | (0.437) | (0.485) |
| RD | 0.599 ** | −0.236 | −0.0187 |
| | (0.259) | (0.371) | (0.401) |
| GDP | 0.004 | 0.0003 | −0.0004 |
| | (0.004) | (0.003) | (0.003) |
| Constant | 78.72 *** | 71.22 *** | 59.11 *** |
| | (3.459) | (3.709) | (4.139) |
| Observations | 902 | 738 | 656 |
| Number | 82 | 82 | 82 |
| R-squared | 0.268 | 0.210 | 0.209 |

Note: *, **, and *** respectively represent that the estimated coefficient is significant at the confidence levels 10%, 5%, and 1%, and the standard error of the coefficient is marked in parentheses.

## 5. Discussion

There are two main views on the impact of environmental regulations on haze concentration. One view is based on "Forced emission reduction", which concludes that environmental regulations have a significant positive effect on energy conservation and emission reduction. Specifically, appropriate environmental regulations can force enterprises to invest in environmental governance and environmental technology innovation, thereby reducing negative environmental impacts. For example, higher environmental standards would restrict the establishment of new enterprises with high pollution emissions and eliminate those with no obvious performance in pollution control, thus improving the environmental quality [51–53]. Another view is based on the "Green paradox", which suggests that environmental regulations not only have no positive impact on the improvement of environmental quality, but also have a negative impact. According to the "Green paradox" theory, improving environmental regulation may make enterprises invest more in environmental management, leading to an increase in costs. At the same time, it also restricts the improvement of production technology and process innovation. This further enables enterprises to increase production and pollution emissions driven by profit maximization goals. In summary, the effect of environmental regulation on haze concentration depends on the magnitude of the effect of positive "Forced emission reduction" and negative "Green paradox". When the positive effect is greater than the negative effect, environmental regulation can significantly contribute to the reduction of haze concentration. Conversely, it acts as a disincentive [54].

Although the fixed-effect model tests for the effect of environmental regulation on haze concentration differed between the five urban agglomerations and non-five urban agglomerations, it is not significant. We use a more accurate threshold model for analysis. Since only five urban agglomerations have passed threshold effect test, this paper analyzed the impact mechanism of environmental regulation on haze concentration and the optimal intensity of environmental regulation in the process of controlling haze in the five urban agglomerations. In the threshold model with five urban agglomerations, the two thresholds are 0.49 and 0.59 respectively. Based on this, environmental regulation is divided into three intervals: low intensity (GER ≤ 0.49), medium intensity (0.49 < GER ≤ 0.59) and high intensity (GER > 0.59). The empirical results of the threshold test show that the impact of environmental regulation on haze concentration (PM2.5) is always significantly negative

in each interval, but the impact degree shows a gradual weakening trend. The effect of regulation (GER) on the environment is lower than that of regulation (49.04); Medium intensity environmental regulation (0.49 < GER ≤ 0.59) has the second effect, while high intensity environmental regulation (GER > 0.59) has the worst effect on haze control, with the absolute value of the coefficient being only 5.23. It further shows that the increase of environmental regulation intensity can significantly reduce the haze concentration of the five urban agglomerations. With the increase of environmental regulation intensity, although the haze concentration always shows a significant decreasing trend, the impact intensity is also weakening. From the theoretical analysis, it can be seen that the positive "Forced emission reduction" of environmental regulation on haze concentration is greater than the negative "Green paradox" in the five major urban agglomerations. As the environmental regulation intensity increases, the positive "Forced emission reduction" effect decreases and the negative "Green paradox" effect increases, but the "Forced emission reduction" effect is always greater than the "Green paradox" effect (Figure 4).

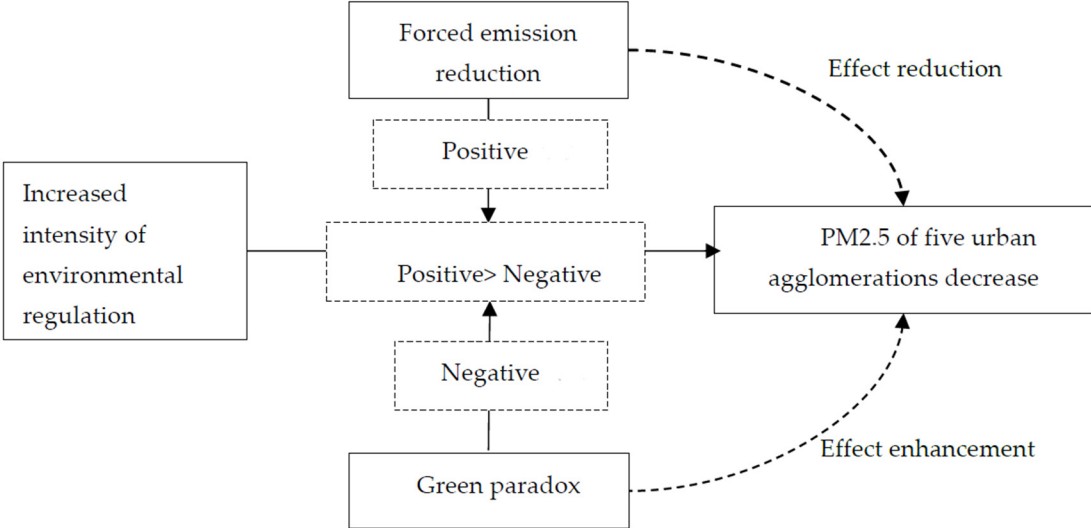

**Figure 4.** Effect of environmental regulation on haze in five urban agglomerations.

What is the most suitable environmental regulation intensity for the five urban agglomerations? In this paper, we speculate based on the results of panel nonlinear model and threshold model. According to the panel nonlinear model, the impact of environmental regulation on haze concentration shows a "U" shape trend, and the threshold is 0.75 ($\frac{-\text{GER}}{-2 \times \text{GER}^2} = \frac{11.64}{2 \times 7.743}$). That is, when the intensity of environmental regulation is less than 0.75, the increase of environmental regulation can reduce the haze concentration. The positive "Forced emission reduction" is greater than the negative "Green paradox". When the intensity of environmental regulation is greater than 0.75, the increase of environmental regulation would increase the concentration of haze, that is, the positive "Forced emission reduction" is smaller than the negative "Green paradox". Although the results are not significant, through the results of the nonlinear model and the range of environmental regulation intensity between [0, 1], we can simply infer that an environmental regulation intensity less than 0.75 is beneficial to haze control. The threshold model further verifies that when the intensity of environmental regulation is less than 0.49, the increase of environmental regulation can significantly reduce the haze concentration, and its impact is the largest in each interval and significant at the 1% level. Therefore, we take the intersection of the results of the two models, that is, environmental regulation is no less than 0.49, which is the most suitable for the five urban agglomerations to control haze.

## 6. Conclusions

Based on the municipal data of 206 cities in China from 2006 to 2016, 82 cities belonging to the 5 urban agglomerations are identified as economically developed regions, while 124 cities outside the 5 urban agglomerations are identified as slightly less economically developed regions. From the perspective of urban agglomeration, this paper attempts to explore whether the impact of environmental regulation on haze intensity can change due to the level of economic development.

The conclusions of this paper are as follows.

(1) In the linear model, increasing environmental regulation can reduce the haze concentration, but this effect is not significant. The effect of environmental regulation on haze control in the five major urban agglomerations is better than that in the non-five major urban agglomerations;

(2) In the nonlinear model, the impact of environmental regulation on haze shows a "U" trend in the five major urban agglomerations and an inverted "U" trend in the non-five major urban agglomerations. By calculating the inflection point, an intensity of environmental regulation less than 0.75 is suitable to reduce haze concentration for the five urban agglomerations. On the contrary, the intensity of environmental regulation greater than 0.54 is suitable to reduce haze concentration for the non-five urban agglomerations. Although the results are not significant, we can still see that the impact of environmental regulation on haze varies depending on the level of regional economic development. Therefore, the intensity of environmental regulation should be formulated according to local conditions;

(3) In the threshold model, there is a threshold effect of environmental regulation on the haze concentration in the five major urban agglomerations, but there is no threshold effect in other cities. In the five major urban agglomerations, although environmental regulation can effectively reduce the haze concentration, the governance effect would weaken as the intensity of environmental regulation increases. When the intensity value of environmental regulation is less than 0.49, the environmental regulation can effectively reduce haze concentration.

Based on the conclusion, this paper puts forward the following policy recommendations.

(1) Establish environmental regulation policies with appropriate intensity. Too strong of a formal environmental regulation policy would restrict the development vitality of enterprises and hinder industrial restructuring. Therefore, we should grasp the strength of formal environmental regulation;

(2) Develop a differentiated and coordinated policy system based on urban agglomeration. There are great differences in the development conditions and institutional environment of China's urban agglomerations. Therefore, when formulating relevant policies, the government should deeply tap into the comparative advantages of different urban agglomerations to develop their strengths and avoid their weaknesses.

Based on 206 Chinese municipal data from 2006 to 2016, including 82 from 5 major urban agglomerations and 124 from non-5 major urban agglomerations, this paper analyzes the impact of environmental regulation on haze concentration and explores whether its impact would vary due to regional economic development. Based on this, we try to find the most suitable environmental regulation intensity for the five major urban agglomerations. Finally, we come to the conclusions, which enrich the theoretical and empirical research in related fields to a certain extent.

Due to the limitation of our ability, further research is needed to solve some unsolved problems and imperfect parts. Firstly, the fixed effect model has not obtained significant results for the five urban agglomerations and non-five urban agglomerations. Although it has obtained linear and nonlinear impact results and verified regional differences, whether the results can help cities reduce haze concentration remains to be further explored in the future. Secondly, due to the difficulties and deficiencies of data collection, this paper only selects the urban data from 2006 to 2016. In the follow-up study, the index calculation

method should be further optimized, and richer data should be collected to further improve the accuracy of the conclusion.

**Author Contributions:** Conceptualization: X.L., C.X. and B.C.; methodology: X.L., J.D. and C.X.; software: X.L.; validation: X.L., W.Z. and B.C.; formal analysis, X.L.; investigation: X.L. and Y.T.; resources: Y.T.; data curation: X.L. and Y.L.; writing—original draft preparation: X.L. and J.D.; writing—review and editing: X.L., W.Z. and Y.L.; visualization: B.C.; supervision: Y.T.; funding acquisition: Y.T. All authors have read and agreed to the published version of the manuscript.

**Funding:** This research was funded by Beijing Social Science Foundation (grant No. 21LLGLC038), Humanities and Social Science Foundation of Ministry of Education of China (grant No. 21YJC630127), Social Science Program of Beijing Municipal Education Commission (grant No. SM202011417010), National Natural Science Foundation of China (grant No. 72073012, 71873016), and Anhui Province University Scientific Research Project (grant No. SK2021A0237).

**Institutional Review Board Statement:** Not applicable.

**Informed Consent Statement:** Not applicable.

**Data Availability Statement:** The data presented in this study are available on request from the corresponding author.

**Acknowledgments:** We are indebted to the anonymous reviewers and the editor.

**Conflicts of Interest:** The authors declare no conflict of interest.

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
