# Peer review of "Can Environmental Regulation Reduce Urban Haze Concentration from the Perspective of China’s Five Urban Agglomerations?"

_atmosphere, doi:10.3390/atmos13050668_

Round 1

Reviewer 1 Report

Li et al. present research and analysis models to verify whether the environmental regulation implemented by local governments can positively influence the decrease in the concentration of haze in different Chinese cities, characterised by different regional heterogeneity.

The overall study seems to be very interesting, but the paper is difficult to read. The article needs a significant improvement in the use of English grammar; several sentences have to be rewritten because they are not clear, some are repeated. I suggest writing the article again and then asking a native English speaker for help.

Talking about the research, as I have already said, it seems interesting, but it is often not well explained: for example, sometimes in the text there are inexplicable numbers, and it is not clear which analysis they come from.

In addition, all acronym must be specified.

Finally, there are also some references that are completely off-topic.

Many of my additional concerns are embedded in the comments below, but I may not catch all of them.

Introduction

  • The introduction is good in terms of content, but almost all sentences need to be rewritten.
  • Choose one of the two sentences: “Therefore, the problem of haze weather is becoming more and more serious and needs to be solved urgently. Therefore, the increasingly serious haze problem needs to be solved.”.

Literature

  • To clarify the three points, maybe it is better to make a list.
  • The second and third points are well quoted, but I suggest changing the order of the quotations.
  • Beware of citations: for example you wrote Greenstone [12], while it is better to write Greenstone et al., or the citation number [15], you cited with the wrong name.
  • The citation [13] and [14] are completely off-topic!!!

Research Model and Method

Data Sources

  • Honestly, I don’t understand how you divided the groups of cities, especially the descriptions of: “82 five major urban agglomerations and 124 non five urban agglomerations”. Please make this important point clearer.
  • You didn’t mention the figure 2.

Model

  • Specify better what GER is.
  • The explanation of the terms of the model needs to be rewritten.
  • I did not understand what the five indices are (maybe they are the five rates indicated later, in case clarify them better) and if the China Urban Statistics Yearbook gives all or only the last one.
  • If you use “i city”, do not change it along the text.
  • What is REGU? Where is he from?
  • Table 1: rewrite more clearly the sentences concerning mainly PEO and ST.

Analysis of Empirical Results

Data Description

  • This paragraph is the same already present in the “Data Sources”. Please rewrite better and in a different way

Panel Fixed Model

  • I do not understand the whole explanation of the U trend. In particular, you said earlier that the inverted “U” trend is shown for the whole country and in the non five major urban agglomerations and then in the latter you have also the U trend. Maybe I missed a step…

Threshold model analysis

  • I did not fully understand the sentence:”Since the implementation and effect of relevant policies of national urban agglomerations are better than those of non-national urban agglomerations, whether the implementation effect of environmental regulation policies on environmental quality is better than that of other regions, resulting in regional differences? “. Please clarify.
  • Maybe the references to table 4 and 5 are wrong.
  • Maybe, you can use “triple threshold” instead of “three threshold”.
  • In this chapter, at one point, you use other GER values, such as 0.72 or 0.83, but I did not understand where to find these values.
  • In the table 5, what are the numbers 1, 2 and 3 next to GEV?

Discussion

  • You wrote “Many foreign scholars…” and you only cited one work. Maybe, it is better to rewrite the sentence in a different way, or to find more works.

Conclusion

  • The first part is completely the same as far as the abstract. I suggest rewriting in a different way.

Reviewer 2 Report

  1. Environmental regulations effectively regulate emissions from point sources and are not so effective from unknown sources. Haze is a phenomenon that occurs due to photochemical reactions, transboundary of pollutants (e.g. dust or biomass burning), and even from the local sources. Since the types and causes of haze are varied and not fully defined in this paper, I’m wondering what assumptions were used in the models to investigate environmental regulations' effectiveness in reducing haze? What types of haze is this study referring to?
  2. In what way the regulation is effective in reducing the haze?
  3. Methodology: The data collection need to be further elaborated.
  4. What aspects of environmental regulations is this paper referring to? If the study aims to investigate the effectiveness of the environmental regulation in reducing haze, I presume the regulations should be related to air pollution, air emissions and air quality. What are theWhat is the relevance of other environmental regulations (see environmental regulation intensity) related to industrial solid waste, domestic sewage treatments, etc.?
  5. All abbreviations (or symbols) in the tables, figures and text should be defined.
  6. The graphs should be appropriately labelled (y-axis)

Reviewer 3 Report

Dear Author

The presented article has valuable insight into the issue of urban enviromental pollution control. The main objective of the study is to build a fixed effect model and a threshold panel regression tecnology to verify the impact of enviromental regulation on smoke haze concentration, and to make clastering tests on five urban agglomerations and non-five urban agglomeration to verify wheter the effect will result in regional heterogeneity due to different levels of regional development. This article is very important to be the basis  for assesing urban urban air quality: Same note are suggested for improvement of this article, as folows: 

A. ABSTRACT

  1. In the anstract, please state the main objectives to be achived.
  2. It is important to emphasize the study recomendations at the and of the abtract

B. INTRIDUCTION

  1. It is very important to emphasize the importance of the study in the introduction to make it easier for the reader to grasp the contents of the entire manuscript.
  2. It is very important to explain the definition of urban agglomeration. For the case of a city like what is the air quality assessed
  3. It is very important to explain the urgency of the study and its contribution to urban development at the end of the introduction.

C. LITERATURE REVIEW

  1. Please add a standard reference that is used as the basis for measuring air quality and for the sake of building a model according to the study result obtained.
  2. Relevant reference support is still needed to be used as a basis for conducting studies and analysis.

D. RESULT AND DISCUSSION

  1. Correlate between urban agglomeration with factors that cause urban air quality degradation based on data obtained in the field.
  2. In what way is the measurement of enviromental degradation carried out in relation to the methodology used.

E. CONCLUSION

  • The conclusion should contain the result and discussion and answer the research abjective to be achieved.
  • Conclusion still need sharpening.
  • What are the weaknesses of this study and what need to be followed up to complement the results of the studies that have been carried out.

Regards,

Good Luck 

Reviewer 4 Report

The key idea driving the research and the arguments presented in this paper is very interesting, i.e. to what extent and how the existing regulation in the field of environment protection influences urban haze concentration. Considering the pace of urbanization and the pressures on the natural environment that it creates, the topic and the focus of the paper are most relevant. That being said, the paper displays substantial weaknesses and these need to be addressed if the paper is to be considered for publication in the journal. My points and suggestions have been outlined beneath:

STYLE ISSUES:

The authors seem to have written the paper, especially the introduction, discussion and conclusions in a great hurry. This might explain repetitions, logical errors, and the like. Consider this:

“The rapid economic development has greatly improved everyone's living standards, but it has also produced a series of external hazards of negative information, such as the increasingly prominent problem of environmental pollution. Among them, haze pollution is the most serious” --  [negative information, such as the increasingly prominent problem of environmental pollution]

“Therefore, the problem of haze weather is becoming more and more serious and needs to be solved urgently. Therefore, the increasingly serious haze problem needs to be solved.”

CONTENT

Section 3.1.

REGULATION INTENSITY? WHAT DO YOU MEAN BY THAT? You mention the term repeatedly, but then it is explained briefly only on page 6. It is necessary that this key for the discussion concept be explained in more detail in the introduction.

Is this what you mean: p 6, Table 1: “The intensity of environmental regulation is calculated by entropy weight method through the five single indexes of industrial SO2 removal rate, smoke and dust removal rate, comprehensive utilization rate of industrial solid waste, domestic sewage treatment rate and harm”

You should also explain to the reader why variation in the regulatory intensity among the selected cities/agglomerations exist. Otherwise, the argument is weak.

It is also important to strengthen the quality of the argument in the introduction, discussion and conclusion. To say that haze urban is a challenge is not enough to justify the development of the model that you do. Please, consider it.

Section 3.2.

Economic quality? – what do you mean by that?

THE SAMPLE:

You argue you chose 206 cities, and yet, you talk about 5 major agglomerations. Please, make the connection clearer to the reader.

Round 2

Reviewer 1 Report

The paper and the corrections remarkably improved the paper. Anyway, it needs more corrections in the English and in the description of the methods and the results, in particular in the abstract and in the conclusions, which are the most important parts of a paper.

Reviewer 3 Report

Dear Author

The improvement in the results was as I expected. Please add in the conclussions section the shortcomings of this study and what needs to be followed up future researchers.

Good luck

Reviewer 4 Report

Thank you for addressing my points and suggestions. I appreciate it. 

Very kind regards, 
